# Gliotoxin Aggravates Experimental Autoimmune Encephalomyelitis by Triggering Neuroinflammation

**DOI:** 10.3390/toxins11080443

**Published:** 2019-07-26

**Authors:** Thais Fernanda de Campos Fraga-Silva, Luiza Ayumi Nishiyama Mimura, Laysla de Campos Toledo Leite, Patrícia Aparecida Borim, Larissa Lumi Watanabe Ishikawa, James Venturini, Maria Sueli Parreira de Arruda, Alexandrina Sartori

**Affiliations:** 1Department of Microbiology and Immunology, Institute of Biosciences of Botucatu, São Paulo State University (UNESP), Botucatu 18618-691, São Paulo, Brazil; 2Department of Biological Sciences, School of Sciences, São Paulo State University (UNESP), Bauru 17033-360, São Paulo, Brazil; 3Department of Tropical Diseases and Image Diagnosis, Botucatu Medical School, São Paulo State University (UNESP), Botucatu 18618-687, São Paulo, Brazil

**Keywords:** multiple sclerosis, experimental autoimmune encephalomyelitis, mycotoxin, gliotoxin, immunomodulation

## Abstract

Gliotoxin (GTX) is the major and the most potent mycotoxin that is secreted by *Aspergillus fumigatus*, which is capable of injuring and killing microglial cells, astrocytes, and oligodendrocytes. During the last years, studies with patients and experimental models of multiple sclerosis (MS), which is an autoimmune disease of the central nervous system (CNS), suggested that fungal infections are among the possible initiators or aggravators of this pathology. The deleterious effect can occur through a direct interaction of the fungus with the CNS or by the toxin release from a non-neurological site. In the present work, we investigated the effect of GTX on experimental autoimmune encephalomyelitis (EAE) development. Female C57BL/6 mice were immunized with myelin oligodendrocyte glycoprotein and then intraperitoneally injected with three doses of GTX (1 mg/kg b.w., each) on days 4, 7, and 10. GTX aggravated clinical symptoms of the disease in a dose-dependent way and this outcome was concomitant with an increased neuroinflammation. CNS analyses revealed that GTX locally increased the relative expression of inflammatory genes and the cytokine production. Our results indicate that GTX administered in a non-neuronal site was able to increase neuroinflammation in EAE. Other mycotoxins could also be deleterious to many neurological diseases by similar mechanisms.

## 1. Introduction

Gliotoxin (GTX) is a mycotoxin that was originally isolated from *Gliocladium* culture. However, it has been lately described that it could also be produced by other fungal species, e.g., *Aspergillus fumigatus*, *Eurotium chevalieri*, *Trichoderma virens*, *Neosartorya pseudofischeri,* and some *Penicillium* and *Acremonium* species [1]. GTX is considered a virulence factor; it intensifies fungal invasiveness [2] and reduces the specific immunity against the fungus [3]. Even though the denomination “gliotoxin” is being applied to both mycotoxins [4] and chemical compounds toxic for glial cells, as 3-chloropropanediol [5], the possible effects of fungi-derived GTX as triggers or aggravators of inflammatory central nervous system (CNS) disorders have not been comprehensively investigated so far.

CNS fungal infections are associated with considerable morbidity and mortality and they comprise a wide spectrum of clinical syndromes, including abscesses, meningitis, meningoencephalitis, stroke, vasculitis, and spinal pathologies, such as arachnoiditis [6]. The main etiological agents of these infections are *Aspergillus*, *Cryptococcus*, *Candida*, *Mucorales*, dematiaceous molds, and dimorphic endemic fungi. The primary routes of infection are respiratory or traumatic inoculation with subsequent hematogenous or contiguous spreading [7]. Certain fungal species that are secluded in non-neuronal tissues can release toxins which can then reach distant tissues, including the CNS, and, in this case, destroy astrocytes and oligodendrocytes [8], which possibly contributes to exacerbate CNS inflammation.

Multiple sclerosis (MS) is an autoimmune disease that is characterized by the presence of autoreactive T cells specific for CNS antigens. Environmental risk factors that are considered to be essential for MS development include infectious and non-infectious factors, as, for example, diet composition, cigarette smoking, and intensity of sunlight exposure [9,10]. Most chronic inflammatory CNS disorders have an infectious origin, and a strong association of viral and bacterial infections with MS was already described [11,12,13]. More recently, Benito-León and Laurence reviewed the possible role of fungal infections in MS development [14]. Several aspects of MS immunopathogenesis, aggravating factors, and therapy approaches have been addressed in experimental models of the disease, such as the experimental autoimmune encephalomyelitis (EAE). The prototypical model is mainly induced in mice by immunization with neuronal antigens emulsified with Freund’s complete adjuvant.

In the present work, we investigated the effect of GTX in EAE development. This GTX was derived from *Gliocladium fimbriatum* and it was administered by intraperitoneal route, i.e., away from the CNS. This approach allowed us to evaluate the ability of GTX, that was administered in a non-neuronal site, to induce inflammation and demyelination in the CNS.

## 2. Results

### 2.1. Gliotoxin Aggravates EAE Evolution

C57BL/6 mice were injected with this toxin at the fourth, seventh, and tenth days after EAE induction (EAE/GTX group) to evaluate the potential of GTX to affect encephalomyelitis development, as indicated by arrows in Figure 1A. The evaluation of the clinical score revealed that the disease onset, which was clinically characterized by paralysis appearance, started between the 9th and 10th days in both EAE control (EAE) and EAE/GTX groups. From the 14th day of the disease on, the EAE/GTX group displayed more pronounced clinical signs in comparison to the EAE group, including a higher clinical score (Figure 1A) and more accentuated body weight loss (Figure 1B). Seventeen days after EAE induction, the EAE/GTX group presented a significantly higher maximum score, which corresponds to the mean of the highest degree of paralysis that was reached by all animals (Figure 1C), associated with a striking loss of body weight (Figure 1D). GTX *per se* neither affected the survival curve nor the liver and kidney function parameters. Contrarily, the injection of GTX in EAE mice caused two deaths in a group with 12 mice, therefore decreasing the animal survival to 83% (Figure 1E). Regarding the liver and kidney function, GTX injection in normal mice (GTX group) did not affect the evaluated clinical biochemical parameters. However, EAE development per se already triggered a reduction in the hepatic enzymes, as shown for ALT (Figure 1F), AST (Figure 1G), and Alk phos (Figure 1H). Urea (Figure 1I) and creatinine (Figure 1J) were also below normal levels. Some of the alterations that were associated with EAE seemed to be reverted by GTX, such as AST, ALT, and creatinine levels.

### 2.2. EAE Aggravation by Gliotoxin Is Dose-Dependent

Considering that the higher dose of GTX (1.0 mg/kg) was able to aggravate EAE but led to a 16.67% of mortality in EAE mice, we next evaluated the effect of lower toxins doses. Two or 10 times smaller GTX doses (0.5 or 0.1 mg/kg) did not significantly interfere in EAE development (Figure 2A,B). Although the mild dose (0.5 mg/kg) increased the total number of leucocytes in the CNS (Figure 2C), only the higher dose of GTX significantly increased IL-17 (Figure 2D) and IL-2 (Figure 2E) production by CNS cell cultures that were stimulated with the neuroantigen (MOG). We next evaluated possible immunological mechanisms involved in this process while taking into account that the highest dose of GTX was able to exacerbate the clinical encephalomyelitis development.

### 2.3. Gliotoxin Triggers Neuroinflammation and Demyelination

The effect of GTX in the CNS was evaluated in lumbar spinal cord samples, seven days after its third dose, in both normal and EAE mice. No inflammation or demyelination was observed in normal mice, as illustrated in Figure 3(A1,B1). However, GTX caused inflammation (Figure 3(A2)) and demyelination (Figure 3(B2)) in this group. As expected, a clear process of inflammation (Figure 3(A3)) and demyelination (Figure 3(B3)) was present in the EAE mice. GTX inoculation in EAE mice resulted in a much more pronounced cell infiltration in the meningeal compartment and cortex (Figure 3(A4)), and demyelination (Figure 3(B4)). Additionally, sodium fluorescein (NaFlu) uptake through the blood-spinal cord barrier was measured to assess the barrier permeability in normal mice, two days after the last GTX dose. Higher NaFlu uptake levels indicated significantly increased permeability in the normal GTX injected mice (Figure 3C).

RT-qPCR analysis of lumbar spinal cord samples revealed that GTX increased the mRNA relative expression of TBX21 (Figure 4A), iNOS (Figure 4C), and ARG1 (Figure 4D) in both GTX and EAE/GTX groups. A significantly elevated expression of TLR2 (Figure 4E) and TLR4 (Figure 4F) was only detected in the EAE/GTX group. In response to MOG, CNS cell cultures from EAE/GTX group produced significantly higher levels of IFN-γ (Figure 4G), IL-17 (Figure 4H), and IL-2 (Figure 4I) in comparison to the EAE group. No production of these cytokines was detected in cell cultures from the CTL and GTX groups that were stimulated with MOG.

### 2.4. Gliotoxin Increases Pro-Inflammatory Cytokine Production by Splenic Cells

The total number of leucocytes in the spleen was similar in the four experimental groups (Figure 5A). However, the production of IL-17 (Figure 5C), IL-2 (Figure 5D), and IL-6 (Figure 5F) was significantly higher in the EAE/GTX group in comparison to the EAE group. Other encephalitogenic cytokines, such as IFN-γ (Figure 5B) and TNF-α (Figure 5E), and the anti-inflammatory cytokine IL-10 (Figure 5G), were similarly elevated in both the EAE and EAE/GTX groups.

## 3. Discussion

The etiology of MS is still unknown, but there is strong evidence that environmental factors, including infections, can trigger or aggravate the disease. Part of the harmful effects of microbes on MS have been attributed to toxins that are derived from bacteria, including staphylococcal, clostridium, and pertussis toxins, and also derived from fungi, such as *Aspergillus* toxins and food-associated mycotoxin ochratoxin A, which is pointed out in a recent review [15]. In the present work, we evaluated the effect of GTX in EAE, which is a murine model to study MS. The choice of GTX was based on the potentially deleterious effect of this mycotoxin due to its neurotoxic properties [4,16,17]. In this scenario, we investigated whether the presence of GTX could exacerbate clinical and histopathological EAE manifestations. Initially, the GTX effect was evaluated in EAE mice that were injected with three doses (1.0 mg/kg) of this mycotoxin. Disease aggravation clearly indicated the occurrence of a deleterious effect. Those animals reached higher clinical scores, lost more body weight, and 16.67% of the mice died. Interestingly, no alterations were observed in biochemical parameters that were associated to liver and kidney dysfunctions in normal mice, i.e., in this specific concentration, the GTX *per se* was not toxic. These findings indicate that the dose employed in our study was far below the toxic doses that were used by other researchers. A single oral dose of GTX (15, 25, or 35 mg/kg), which are at least 15 times higher than our working dose can cause death in hamsters within 12 h, and the surviving animals present hepatic alterations [18]. It was also observed that, in association with other conditions, the intraperitoneal injection of GTX (100 μg/mouse) results in 25% of death in mice immunosuppressed by sublethal irradiation, but not in normal mice; however, in higher doses (150 and 200 μg/mouse), GTX causes death in both immunosuppressed and normal mice [19]. Another study showed that a single intraperitoneal injection of GTX (3.0 mg/kg) in rats does not affect the viability of hepatocytes in normal liver, but rather triggers a greater number of apoptotic hepatocytes in cirrhotic liver; the authors suggested that activated cells are more sensitive to GTX [20].

When considering the evidence from the abovementioned literature, which leads us to believe that the deleterious effect is dose dependent, we evaluated the effect of lower doses of GTX on EAE development. We initially chose a GTX dose (1.0 mg/kg) that was based on studies that use intraperitoneal route administration more than once. To prevent diabetes mellitus in rats, for example, GTX was weekly administered three times during chronic treatment [21], and to reduce colitis in rats, GTX was given twice a day during seven or 21 days for acute and chronic treatments, respectively [22]. Regarding the time points for mice treatment with the mycotoxin, we chose to inject GTX on days four, seven, and 10 after EAE induction. This choice was based on disease immunopathogenesis, i.e., the first dose during the innate immune response and the second and third ones during the activation/induction of the adaptive immune response. Our results are in agreement with the literature, since the effect of GTX on clinical and immunological parameters of EAE were only observed with the highest dose (1.0 mg/kg). Although the mild dose (0.5 mg/kg) significantly increased the total number of cells in the CNS, it did not affect the disease clinical score; the lowest dose (0.1 mg/kg) also did not affect the clinical score or the immunological parameters. Therefore, the highest dose was used to address the possible mechanisms that are involved in disease aggravation.

Further CNS evaluation allowed to confirm the higher disease severity in GTX treated mice. Histopathological analysis indicated a more accentuated degree of inflammation and demyelination in the lumbar spinal cord of EAE mice that were treated with GTX. These alterations have been frequently displayed in lumbar spinal cord of EAE mice, and they are considered as disease hallmarks [23]. As indicated by the histopathological analysis, a more intense inflammatory process was clearly co-localized, with more accentuated demyelinating lesions. This is interesting, because demyelination intensity has been an indication of disease severity in both EAE and MS [24,25]. The more intense demyelination found in the EAE/GTX group could be a direct consequence of GTX neurotoxic effect, while considering that GTX administration in normal mice resulted in discreet inflammatory demyelinating lesions. Literature reports reinforce the possibility that GTX plays a direct neurotoxic effect. For example, the local injection of chemical GTX damages astrocytes, endothelial cells, oligodendrocytes, and their precursors, and the induction of oligodendrocyte apoptosis, results in rapid local demyelination and localized activation of microglial cells [26]. Additionally, it was recently described that GTX derived from *G. fimbriatum*, the same mycotoxin used in the present study, penetrates and impairs the integrity of the human blood-brain barrier in vitro [27]. In this sense, in this work we also demonstrated that GTX enhanced the permeability of the blood-spinal cord barrier in normal mice and this increased permeability allowed for inflammation and demyelination in the CNS.

The CNS immunological parameters evaluation confirmed the existence of a higher degree of inflammation. The mRNA expression performed in lumbar spinal cord samples revealed that GTX increased T-box transcription factor (TBX21), inducible nitric oxide synthase (iNOS), and arginase 1 (ARG1) in both normal and EAE mice. These results highly suggest that GTX is capable to determine inflammation in the CNS, possibly by facilitating the access of pro-inflammatory cells, since TBX21 and iNOS/ARG1, which are Th1 cells subset signature and common macrophage transcription markers, respectively [28,29], were significantly increased in GTX and EAE/GTX groups. Intriguingly, the EAE/GTX group showed increased expression of toll-like receptors (TLR) 2 and 4 in comparison to the other groups. Although no evidences are available regarding a direct relationship between GTX and TLRs, a recent review discusses the possible effects of microbial metabolites on TLR expression in the context of autoimmune diseases. According to the authors, microbial toxins elicit the expression of TLRs and pro-inflammatory mediators that upregulate and activate tissue damage, leading to the aggregation of damage-associated molecular patterns that can activate immune cells and result in chronic inflammation and autoimmunity [30].

Additionally, we evaluated whether CNS eluted cells from GTX treated mice produced higher levels of encephalitogenic cytokines in response to the neuroantigen. Normal mice, whether treated or not with GTX, did not produce cytokines in response to MOG. Nevertheless, cells from the EAE mice treated with GTX produced significantly higher levels of IFN-γ, IL-17, and IL-2 in comparison to cells from EAE mice that were not treated with GTX. It is well established that these cytokines play a decisive neuroinflammatory role in the pathogenesis of both EAE and MS; they are the main soluble mediators of Th1 and Th17 phenotypes [31,32,33]. Quantification of inflammatory mediators of mRNA expression by PCR in lumbar spinal cord samples and the cytokines from CNS cells cultures confirmed and reinforced the finding that GTX increases neuroinflammation, aggravating the autoimmune response in EAE.

Peripheral cytokine production in response to MOG was next checked in spleen cell cultures to unravel a possible systemic deleterious effect of this mycotoxin. Although the total splenic cell number was similar among the four experimental groups, the cytokine production partially resembled CNS production. Encephalitogenic cytokines, such as IL-17, IL-2, and IL-6, were clearly upregulated in the spleen cell cultures from EAE/GTX mice. Intriguingly, these findings were in contrast to the massive description of GTX immunosuppressive effect observed in in vitro experiments with innate and adaptive immune cells [3,34,35,36,37,38,39]. When considering that the GTX effect was dose dependent, and also that cell cultures were performed seven days after the last GTX dose, it is possible that a restoration of the homeostasis of the immune system has occurred during this time period. Different approaches have been studied to restore the immune system, particularly in immunocompromised individuals. A weaker immune system can, for example, be stimulated to provide a response if the stimulus is considerably strengthened and enhanced [40]. This phenomenon seems to occur in our disease model, in which the neuroantigen is strongly available and persistent.

The belief that fungi play an important role in MS is emerging [14,15]. MS resolution by antifungal therapy [41,42] comprises a strong evidence that supports fungi major contribution to this disease. According to Purzycki and Shain [8], a compelling connection between fungal toxins and MS is becoming more evident. According to these authors, toxins that are released by certain pathogenic fungi, such as species of *Candida* and *Aspergillus* located in non-neuronal tissues, can damage astrocytes and oligodendrocytes, which triggering the release of myelin antigens. Altogether, our results experimentally demonstrated that gliotoxin triggers inflammation and aggravates encephalomyelitis, which reinforces the correlation between fungal infections or its metabolites and CNS neurodegenerative disease as MS.

## 4. Materials and Methods

### 4.1. Animals

Female C57BL/6 mice (*Mus musculus*) with 9–11 weeks old were purchased from the University of São Paulo (USP) (Ribeirão Preto, SP, Brazil). The mice were allocated in specific-pathogen free conditions, in cages (maximum 5 mice per cage) with free food and autoclaved filtered water in a controlled photoperiod (12h/12h, dark/light cycle) environment. The animals were manipulated according to the ethical principles for animal research that were adopted by the National Council for the Control of Animal Experimentation. This study was approved by the local Ethics Committee for Animal Experimentation, São Paulo State University (UNESP) (Botucatu, SP, Brazil; protocol number 351, approved on 30 November 2011).

### 4.2. EAE Induction

The mice were subcutaneously immunized with 25 μL (100 μg) of MOG_35–55_ peptide (MEVGWYRSPFSRVVHLYRNGK, Genemed Synthesis Inc., San Antonio, TX, USA), emulsified in 25 μL of Complete Freund’s Adjuvant (Sigma-Aldrich Corporation, St. Louis, MO, USA) containing 4 mg/mL of *Mycobacterium tuberculosis*. Mice also received two intraperitoneal doses of 200 ng of *Bordetella pertussis* toxin (Sigma) at 0 and 48 h after immunization. Clinical score and body weight were daily recorded until the 17th day. EAE clinical scores were monitored according to the following criteria: 0—no symptoms; 1—limp tail; 2—hind legs weakness; 3—partially paralyzed hind legs; 4—complete hind leg paralysis; and, 5—complete paralysis.

### 4.3. Fungal Toxin and Experimental Design

Gliotoxin (GTX) from *Gliocladium fimbriatum* (Merck Millipore Corporation, Darmstadt, Germany) was initially diluted in dimethyl sulfoxide solution (DMSO, 1.0 mg/mL, ≥99.5% GC, Sigma) and then adjusted with 0.9% sterile saline solution (SSS) to be intraperitoneally injected in mice (1.0 mg/kg). The animals were allocated into four groups: CTL, injected with three doses of DMSO/SSS; GTX, injected with three doses of GTX; EAE, submitted to EAE induction and injected with three doses of DMSO/SSS after EAE induction; and EAE/GTX, injected with three doses of GTX after EAE induction. All of the inoculations were performed at the fourth, seventh, and tenth days after the beginning of the experiment (day 0). Experimental evaluations were carried out at the acute disease phase (day 17).

### 4.4. Hepatic and Renal Function

Concentrations of alanine aminotransferase (ALT), aspartate aminotransferase (AST), alkaline phosphatase (AF), urea, and creatinine serum levels were quantified with Bioclin commercial kits (Quibasa Química Básica Ltda., Belo Horizonte, MG, Brazil). Results were measured by Cobas Mira plus Chemistry Analyzer (Roche Diagnostics, Basel, Switzerland).

### 4.5. Histopathology

The histological analysis was performed during clinical EAE peak phase, i.e., 17 days after disease induction. After euthanasia, lumbar spinal cord samples were removed and then fixed in 10% neutral buffered formalin. Paraffin slides with 4 μm were stained with hematoxylin and eosin (H & E) to assess cell infiltration and with luxol fast blue (LFB) to assess demyelination. All slides were analyzed in a Nikon microscope (Nikon Corporation, Melville, NY, USA).

### 4.6. Blood–Spinal Cord Barrier Permeability Assay

Spinal cord barrier permeability was tested by the sodium fluorescein (NaFlu) assay (Christy et al., 2013). Two days after the last GTX dose, the mice were intraperitoneally inoculated with 100 μL of NaFlu (Sigma Aldrich). After 20 min., mice were anesthetized with ketamine/xylazine, blood samples were collected by cardiac puncture with heparinized syringe, and mice were perfused with 10 mL of SSS. The spinal cord was collected and homogenized with 400 μL of SSS. After centrifugation at 9000 rpm for 10 min. at 22 °C, the supernatants and the plasma samples were transferred to black 96-well immuno plates. Fluorescence was measured in BioTek Synergy™ microplate reader (BioTek Instruments, Inc., Winooski, VT, USA), emission 485 nm/excitement 528 nm. The results were expressed in relative fluorescence units (RFUs). The following equation was used to evaluate NaFlu uptake: (sample RFU/sample weight)/(plasma RFU/blood volume).

### 4.7. RT-qPCR Analysis

RNA from the frozen lumbar spinal cord samples was extracted with the TRIzol reagent (Life Technologies, Carlsbad, CA, USA) and 1000 ng of RNA was converted to cDNA while using High Capacity cDNA Reverse Transcription kit (Life Technologies). Expression of TBX21 (Mm00450960_m1), RORc (Mm01261022_m1), NOS2 (Mm00440502_m1), ARG1 (Mm00475988_m1), TLR2 (Mm00442346_m1), and TLR4 (Mm00445273_m1) target genes were analyzed and normalized with GAPDH (Mm99999915_m1). Real Time PCR was performed while using TaqMan™ Gene Expression Assays (Applied Biosystems, Foster City, CA, USA) in an ABI 7300 equipment (Applied Biosystems). Data were analyzed in SDS Software System 7300 and the relative quantification was determined based on fold difference (2^−ΔΔCt^) while using the Ct value of the target gene normalized to the reference gene, and the control group (normal mice) as the calibrator.

### 4.8. CNS-Mononuclear Cells Isolation

Seventeen days after EAE induction, the mice were anesthetized with ketamine/xylazine and perfused with 10 mL of SSS. Brain and spinal cord (CNS samples) were collected and digested with 2.5 mg/mL of collagenase D (Roche Applied Science, Indianapolis, IN, USA) at 37 °C for 45 min. Suspensions were washed in RPMI-1640 medium (Sigma Aldrich), centrifuged at 450× *g* and 4 °C for 15 min., resuspended in a 30% Percoll gradient (GE Healthcare, Uppsala, Sweden), and then gently laid over a 70% Percoll gradient. The tubes were centrifuged at 950× *g* for 20 min., with the centrifuge breaks turned off. The interface between the two Percoll gradients containing the mononuclear cells was collected and resuspended in supplemented RPMI medium (1% gentamicin, 2% glutamine, 1% sodium pyruvate, 1% non-essential amino acids, and 10% fetal bovine serum). The cells from two animals were pooled to perform the experiments.

### 4.9. Cell Culture Conditions and Cytokine Quantification

Spleen cells were collected, lysed with buffer containing NH_4_Cl, and adjusted to 5 × 10^6^ cells/mL in complete RPMI medium (1% gentamicin, 2% glutamine, and 10% fetal bovine serum). CNS mononuclear cells were adjusted to 2 × 10^5^ cells/mL in supplemented RPMI medium. The spleen and CNS cells were plated and stimulated with MOG (20 μg/mL and 50 μg/mL, respectively). Cytokine levels were evaluated 48 h later by enzyme-linked immunosorbent assay (ELISA) in culture supernatants by using IFN-γ and IL-10 BD OptEIA Sets (Becton, Dickinson and Company, BD, Franklin, San Diego, CA, USA), and IL-6, IL-17, and TNF-α Duosets (R&D Systems, Minneapolis, MN, USA). The assays were performed according to the manufacturer’s instructions.

### 4.10. Statistical Analysis

The results are expressed as mean ± standard error deviation (SEM). Comparisons between two groups were made by t test and One-Way ANOVA, followed by Tukey’s test for parametric variables, made comparisons among three or more groups. The GraphPad Prism v5.0 Statistical Guide (2007, GraphPad Software Inc., San Diego, CA, USA) for Windows was used to analyze the data and create the graphs.

## Figures and Tables

**Figure 1 toxins-11-00443-f001:**
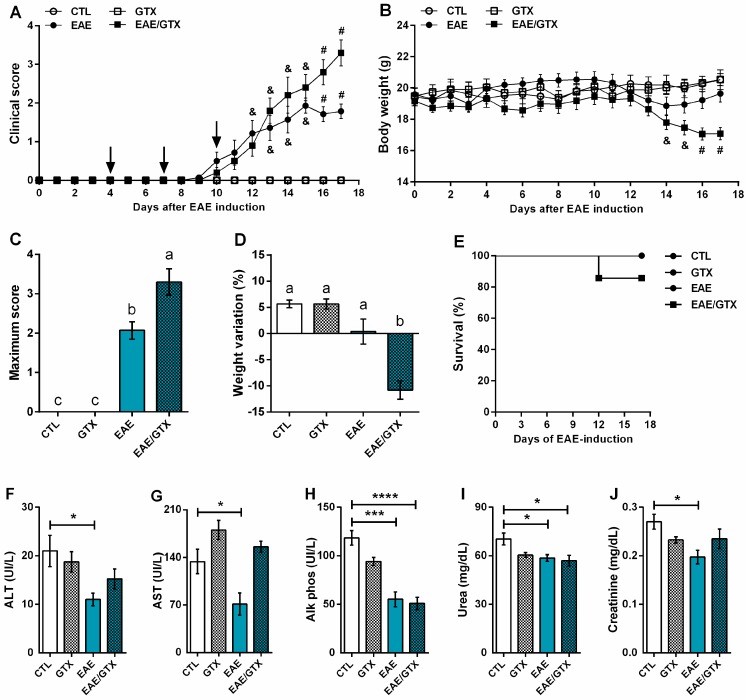
GTX aggravates encephalomyelitis (EAE) development. Normal and EAE mice received three GTX doses (1.0 mg/kg). Clinical score (**A**) and body weight (**B**) were daily analyzed until the acute disease phase (day 17). Maximum clinical score (**C**), percentage of weight variation (**D**), and percentage of survival (**E**) were determined during EAE-development, i.e., from day 0 to day 17. Biochemical parameters as alanine aminotransferase (ALT) (**F**), aspartate aminotransferase (AST) (**G**), Alk phos (**H**), urea (**I**), and creatinine (**J**) were analyzed on day 17 after EAE induction. From A to E, the results are representative of three independent experiments and the statistical analysis was performed among groups (*n* = 10–14); (**A**,**B**) & *p* < 0.01 vs. CTL and GTX groups and # *p* < 0.001 vs. other groups; (**C**,**D**) distinct letters indicate statistical difference among groups, *p* < 0.001. From F to J the results are representative of one experiment and the statistical analyses was performed between control (CTL) (*n* = 3) vs. other groups (*n* = 4), * *p* < 0.05, *** *p* < 0.001 and **** *p* < 0.0001.

**Figure 2 toxins-11-00443-f002:**
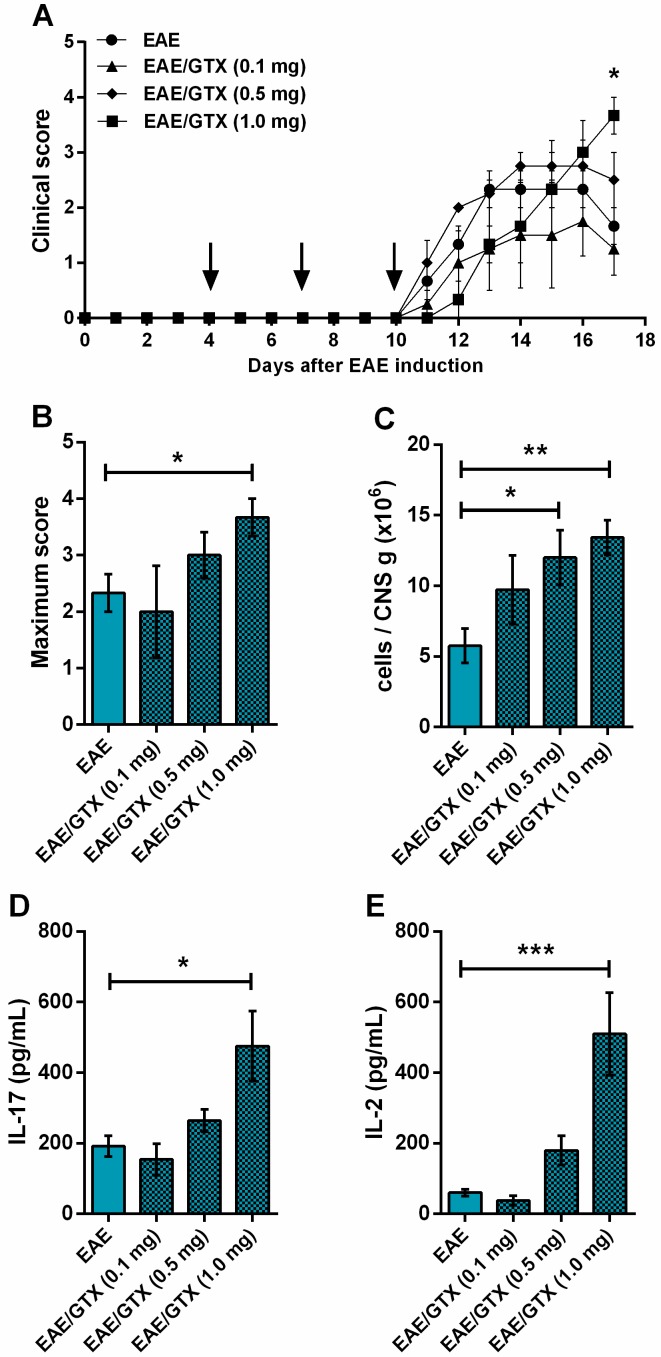
Dose-dependent effect of GTX on EAE aggravation. Mice received three GTX doses after EAE induction. Each group was injected with different concentrations (0.1, 0.5, or 1.0 mg/kg). Clinical scores (**A**) were daily analyzed until the acute disease phase (day 17). Maximum clinical score (**B**) was determined during EAE development, i.e., from day 0 to day 17. Total leukocyte number per gram of central nervous system (CNS) tissue (**C**), and cytokine production as IL-17 (**D**) and IL-2 (**E**) were analyzed on day 17 after EAE induction. The results are representative of two independent experiments and the statistical analysis was performed between EAE (*n* = 6) vs. other groups (*n* = 4–6), * *p* < 0.05, ** *p* < 0.01 and *** *p* < 0.001.

**Figure 3 toxins-11-00443-f003:**
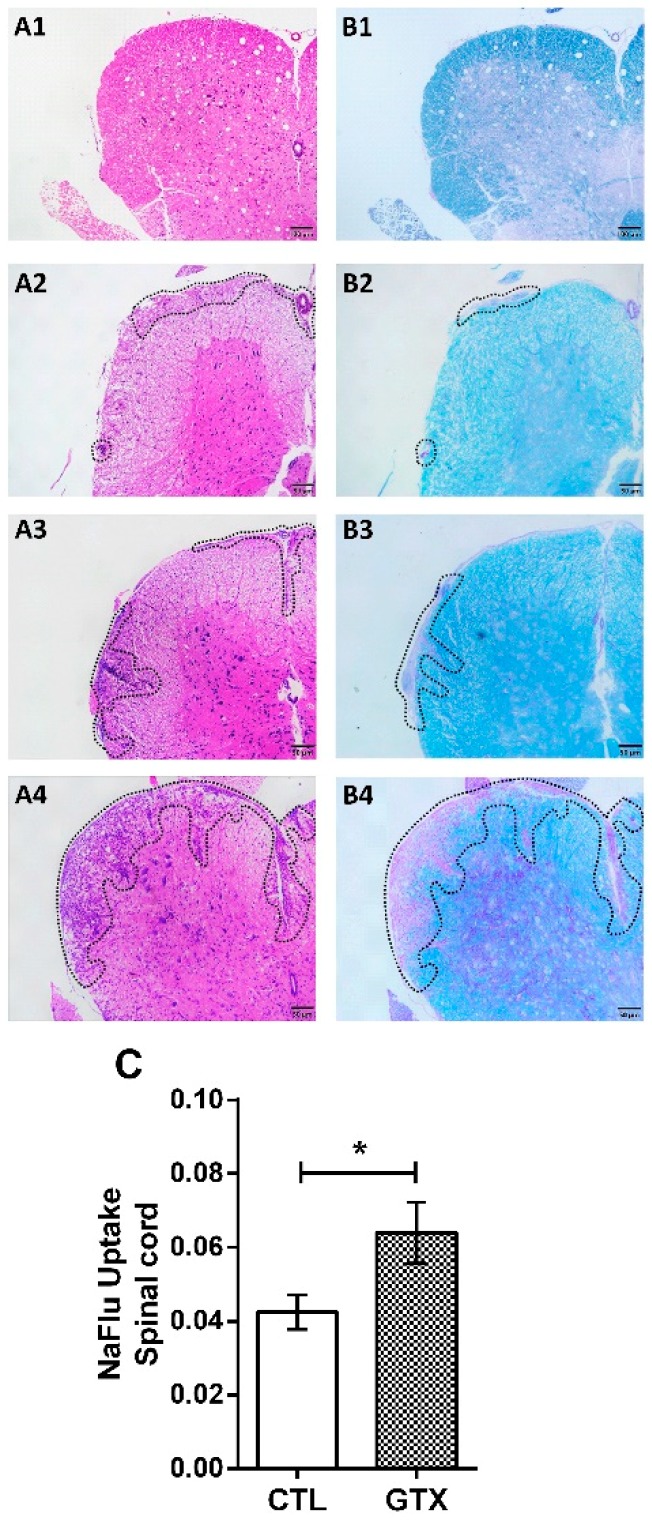
GTX enhances neuroinflammation and demyelination in EAE mice. Normal and EAE mice received three GTX doses (1.0 mg/kg). Inflammatory infiltration was evaluated in hematoxylin/eosin (**A**) and demyelination in luxol fast blue (**B**) stained sections of control (CTL) (**1**), gliotoxin (GTX) (**2**), EAE (**3**), and EAE/GTX (**4**) groups. The blood-spinal cord barrier permeability test (NaFlu uptake) was assessed in spinal cord homogenates obtained from non-EAE mice (**C**). NaFlu uptake results are representative of one experiment and the statistical analysis was performed between CTL vs. GTX groups (*n* = 6), * *p* < 0.05.

**Figure 4 toxins-11-00443-f004:**
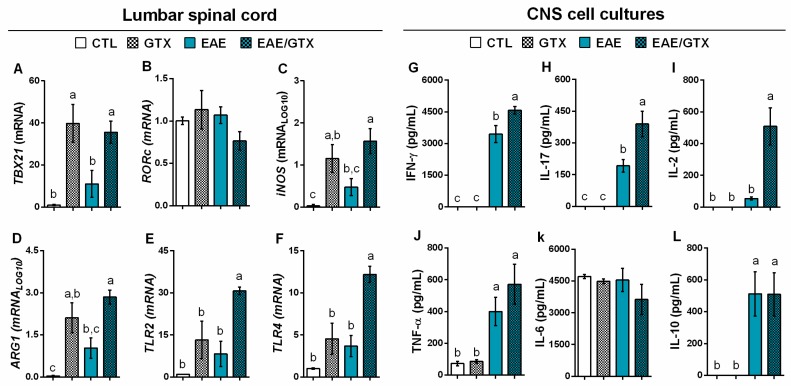
GTX triggers a highly pro-inflammatory environment at the CNS. Normal and EAE mice received three GTX doses (1.0 mg/kg) and CNS analyses were performed at the acute disease phase (day 17). The mRNA relative expression of T-box transcription factor (TBX21) (**A**), RAR related orphan receptor C (RORc) (**B**), inducible nitric oxide synthase (iNOS) (**C**), arginase 1 (ARG1) (**D**), TLR2 (**E**), and TLR4 (**F**) were assessed in lumbar spinal cord samples. The production of IFN-γ (**G**), IL-17 (**H**), IL-2 (**I**), TNF-α (**J**), IL-6 (**K**), and IL-10 (**L**) were quantified in the supernatant of CNS cell cultures (5 × 10^5^ cells/mL) stimulated with neuroantigen (MOG) (50 µg/mL). The results are representative of two independent experiments and the statistical analysis was performed among groups (*n* = 3–6); distinct letters indicate statistical differences among groups, (A,C,J and L) *p* < 0.01 and (D,E,F,G,H and I) *p* < 0.001.

**Figure 5 toxins-11-00443-f005:**
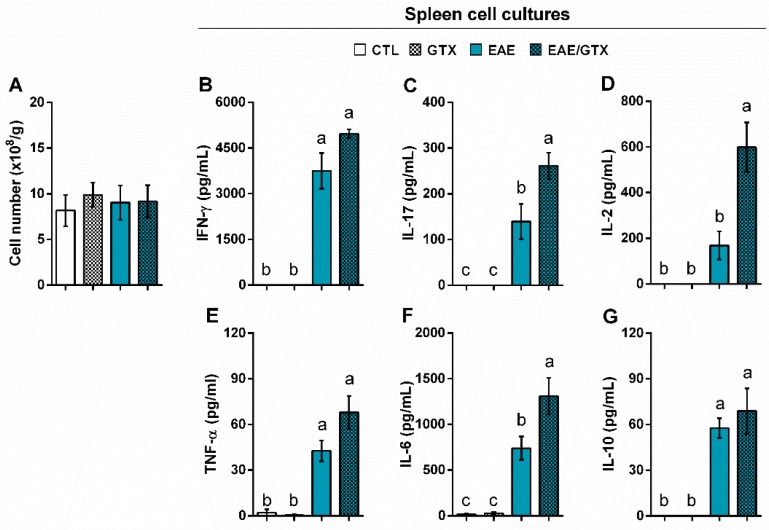
GTX increases splenic cytokine production in EAE mice. Normal and EAE mice received three GTX doses (1.0 mg/kg) and spleen analyses were performed at the acute disease phase (day 17). Total leukocyte number per gram of spleen (**A**) and production of IFN-γ (**B**), IL-17 (**C**), IL-2 (**D**), TNF-α (**E**), IL-6 (**F**), and IL-10 (**G**) by spleen cell cultures (5 × 10^6^ cells/mL) stimulated with MOG (20 µg/mL). The results are representative of two independent experiments and the statistical analysis was performed among groups (*n* = 3–6); distinct letters indicate statistical differences among groups, (**B**–**G**) *p* < 0.001.

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
