# Peer review of "Gliotoxin Aggravates Experimental Autoimmune Encephalomyelitis by Triggering Neuroinflammation"

_toxins, 2019, doi:10.3390/toxins11080443_

Round 1

Reviewer 1 Report

The authors have addressed reviewers' comments, included additional data and revised the manuscript accordingly. Therefore, I would like to recommend the current version for publication.

Author Response

The authors are very grateful for the review. One of the authors is currently living in the USA and had proper training for the manuscript proofreading. Minor changes were made to improve clarity and conciseness for a better understanding of the authors’ rationale.

Reviewer 2 Report

Dear Authors here are my comments and suggestions:

In the Abstract please explain MOG35-55.   

In the section 2. Results please explain to what exactly clinical score and maximum score stand for. 

In figure 1. A-J explain abbreviation CTL is not explained; I suggest to use control instead

Figure 1. C and D- what is a, b, c?

Line 87: Use (n = 10-14) instead of "... whose n= 10-14

Line 89 and 90: the exact value of p should be written

In the section 2.2 EAE aggravation by GTX is dose-dependent  figure 2C is not cited- please correct and include in the text. 

In each graph of the figure 2 please specify the exact dose of GTX using parentheses as it would better match with the text section.

Line 98 and 99: the exact value of p should be written

Figure 3 should be re-labeled. Since neuroinflammation and demyelination are the evaluated effects these could be labeled as A and B, while the applied treatments may be labeled as 1, 2,3 and 4. After the labels are corrected please cite in the text (section 2.3. Gliotoxin triggers neuroinflammation and demyelination) following order of the labels- it would read more easily.

I suggest that Figure 3I is labeled as new figure 4. Please briefly explain NaFlu uptake meaning in the text (section 2.3. Gliotoxin triggers neuroinflammation and demyelination).

In Figure 4 please add explanation for statistical differences with the exact p value

In line 135: Use (n = ....) instead of "... whose n= ...

In Figure 5 please add explanation for statistical differences with the exact p value

In line 143: Use (n = ....) instead of "... whose n= ...

In the section 3. Discussion line 152: instead of "we asked..." please re-write to something like we investigated or explored

In the section 4.3. Fungal toxin and experimental design dimethyl sulfoxide (DMSO) should be written, including its grade, purity, manufacturer

In the section References: Please correct the spaces and correct according to the journal  instructions

Author Response

We are very grateful for all suggestions. Please find below the answers to each of the addressed issues. The all the changes made in response to the reviewer 2 are highlighted in yellow throughout the manuscript.

In the Abstract please explain MOG35-55.  

R.: As suggested, MOG35-55 was changed for myelin oligodendrocyte glycoprotein in the Abstract section.

In the section 2. Results please explain to what exactly clinical score and maximum score stand for.

R.: As recommended, we modified the text to clarify the meaning of clinical score and maximum score in the Results section and, also, in the Figure 1 legend. 

In figure 1. A-J explain abbreviation CTL is not explained; I suggest to use control instead R.: As suggested, the meaning of abbreviation CTL was explained in Figure 1 legend.

Figure 1. C and D- what is a, b, c?

R.: Distinct letters indicate statistical difference among groups and this information and the exact p value of ANOVA test was added in the legends of figures.

Line 87: Use (n = 10-14) instead of "... whose n= 10-14 ok line 91

In line 135: Use (n = ....) instead of "... whose n= ...

In line 143: Use (n = ....) instead of "... whose n= ...

R.: We agree with the reviewer and applied the suggestions to all figures.

Line 89 and 90: the exact value of p should be written.

Line 98 and 99: the exact value of p should be written.

R.: The exact p value was added in the legends of Figures 1 and 2.

In the section 2.2 EAE aggravation by GTX is dose-dependent figure 2C is not cited- please correct and include in the text.

R.: Figure 2C was cited in the text.

In each graph of the figure 2 please specify the exact dose of GTX using parentheses as it would better match with the text section.

R.: We agree with the reviewer and we specified the exact dose of GTX using parentheses in Figure 2, in both Results and Discussion sections.

Figure 3 should be re-labeled. Since neuroinflammation and demyelination are the evaluated effects these could be labeled as A and B, while the applied treatments may be labeled as 1, 2,3 and 4. After the labels are corrected please cite in the text (section 2.3. Gliotoxin triggers neuroinflammation and demyelination) following order of the labels- it would read more easily.

R.: We entirely agree and accepted the suggestion. Figure 3 was re-labeled and the corrected citation in the text was done.

I suggest that Figure 3I is labeled as new figure 4. Please briefly explain NaFlu uptake meaning in the text (section 2.3. Gliotoxin triggers neuroinflammation and demyelination).

R.: We would prefer to keep the NaFlu test as an integral part of Figure 3 because it clear shows an direct relationship between increased blood-spinal cord permeability and increased inflammation/demyelination. As suggested, NaFlu uptake was explain in Results (section 2.3).

In Figure 4 please add explanation for statistical differences with the exact p value.

In Figure 5 please add explanation for statistical differences with the exact p value.

R.: For each graph an ANOVA test was performed and each presented a distinct value of p. We would prefer to describe the differences with letters to avoid visually pollution. Distinct letters indicate statistical difference among groups and this information and the exact p value of ANOVA test was added in the legends of figures.

In the section 3. Discussion line 152: instead of "we asked..." please re-write to something like we investigated or explored.

R.: We appreciated this suggestion and re-write using investigated. 

In the section 4.3. Fungal toxin and experimental design dimethyl sulfoxide (DMSO) should be written, including its grade, purity, manufacturer.

R.: All information was included in Materials and Methods (section 4.3).

In the section References: Please correct the spaces and correct according to the journal  instructions.

R.: We used Zotero to organize the references, with “Multidisciplinary Digital Publishing Institute” style as indicated in Toxins — Instructions for Authors.